# Bioremediation of Heavy Metals by the Genus Bacillus

**DOI:** 10.3390/ijerph20064964

**Published:** 2023-03-11

**Authors:** Monika Wróbel, Wojciech Śliwakowski, Paweł Kowalczyk, Karol Kramkowski, Jakub Dobrzyński

**Affiliations:** 1Faculty of Biology, University of Warsaw, Miecznikowa 1, 02-096 Warsaw, Poland; 2Institute of Technology and Life Sciences—National Research Institute, Falenty, 3 Hrabska Avenue, 05-090 Raszyn, Poland; 3Department of Animal Nutrition, The Kielanowski Institute of Animal Physiology and Nutrition, Polish Academy of Sciences, Instytucka 3, 05-110 Jabłonna, Poland; 4Department of Physical Chemistry, Medical University of Białystok, Kilińskiego 1 Str., 15-089 Białystok, Poland

**Keywords:** biological removal of heavy metals, spore-forming bacteria, sustainable environmental management

## Abstract

Environmental contamination with heavy metals is one of the major problems caused by human activity. Bioremediation is an effective and eco-friendly approach that can reduce heavy metal contamination in the environment. Bioremediation agents include bacteria of the genus *Bacillus*, among others. The best-described species in terms of the bioremediation potential of *Bacillus* spp. Are *B. subtilis*, *B. cereus*, or *B. thuringiensis*. This bacterial genus has several bioremediation strategies, including biosorption, extracellular polymeric substance (EPS)-mediated biosorption, bioaccumulation, or bioprecipitation. Due to the above-mentioned strategies, *Bacillus* spp. strains can reduce the amounts of metals such as lead, cadmium, mercury, chromium, arsenic or nickel in the environment. Moreover, strains of the genus *Bacillus* can also assist phytoremediation by stimulating plant growth and bioaccumulation of heavy metals in the soil. Therefore, *Bacillus* spp. is one of the best sustainable solutions for reducing heavy metals from various environments, especially soil.

## 1. Introduction

Heavy metals are a collection of metals and semi-metals, characterized by high density and usually toxic properties [1,2,3,4,5]. Of all the heavy metals found in the environment, dangerously increased amounts of Cd and Pb are the biggest concern, i.e., ballast elements that are completely unnecessary for living organisms [1]. These metals enter the biological cycle to the greatest extent through crops, taking up metals from soils [1,6,7,8]. Studies have shown that these elements caused changes in the cell cycle, carcinogenesis, or apoptosis [9].

In 2015, the United Nations (UN) set a key sustainable development objective to reduce diseases and deaths associated with soil contamination by 2030 [10]. To achieve this goal, it is necessary to seek sustainable methods for remediating heavy metals in soil [11]. There are several conventional techniques to remove heavy metals including chemical precipitation, oxidation or reduction, filtration, ion exchange, reverse osmosis, membrane technology, evaporation, and electrochemical treatment. However, most of these techniques are becoming ineffective [8,12,13]. Metals in soils form such stable compounds, that natural removal processes are unable to remove them [3,4,6,14,15]. Therefore, it is extremely difficult to reduce the influx of these toxic compounds into the human body [1]. Thanks to modern research, the now-improved bioremediation methods using suitable microbial species (that can act alone or support the action of hyperaccumulators) are becoming more common in environmental protection [8,16].

One way of bioremediation may be the use of bacteria of the genus *Bacillus* [8,17,18,19,20,21]. They are Gram-positive, spore-forming, rod-shaped, and aerobic or facultative anaerobes. Overall, the genus *Bacillus* is most commonly found in soil, but can also be isolated from other sources, e.g., water, air, water, vegetables, and food, as well as human and animal intestines [22,23,24,25,26,27]. The unique trait of *Bacillus* spp. is the ability of spore-forming under extreme conditions. Due to their specific structure, the spores are able to resist significant environmental stresses, including high temperature, drought, humidity, and radiation. This characteristic gives them an advantage over other bacteria and makes them eagerly used commercially in various fields of industry and agriculture [8,28,29,30,31,32,33,34,35,36,37].

Thus, the review aims to summarize the current knowledge of the possibility of using *Bacillus* spp. directly in soil bioremediation, and in supporting phytoremediation, a technology that uses higher plants in the environmental clean-up processes.

## 2. Scale of Heavy Metal Contamination

Heavy metals present in the environment are of various origins. They can be natural processes, such as rock weathering, volcanic eruption, forest fires, or soil-forming processes. However, the most significant sources of heavy metal contamination are anthropogenic processes [1,2,15].

Metals have been, and continue to be, important raw materials for economic development [3,5]. Since the early Middle Ages, numerous mines have been established in Europe in areas where metal ores were shallow, i.e., silver and lead, gold, arsenic and gold, copper, tin and iron [38]. Nevertheless, their extraction and processing contribute to strong local contamination of the environment, especially the soil [5,6]. Particularly high concentrations of metals are associated with waste resulting from the historical processing of sulfide metal ores [5]. As mentioned previously, nowadays, metals have application in numerous industries and the volume of emissions resulting from their processing strongly varies. Currently, the dominant source of atmospheric emissions of most heavy metals is the stationary combustion of solid and liquid fuels in the power industry, accounting for more than a half of total emissions from anthropogenic sources [5,38,39].

In Europe, there are approximately 2.5 million sites potentially contaminated with heavy metals and organic pollutants [40]. For the US, between 235 000 and 355 000 sites require remediation [11]. Estimating the total degraded area globally is not easy. It is reported to range from less than 1 to more than 6 billion hectares, with widespread disagreement on spatial distribution [3].

In some cases, regulations on the permissible amounts of heavy metals in soil vary from country to country. For example: the highest permissible amount of Pb in soil in Romania is 50 mg kg^−1^, while in the Netherlands it is 140 mg kg^−1^. In contrast, for Cr, the maximum amount allowed in soil for both countries is the same: 100 mg kg^−1^ [2]. Results from the literature indicate that in many countries, the amounts of some heavy metals present in soil far exceed the permissible amounts. For example, in Iran, the amount of Pb in some contaminated soils has been measured at 57 mg kg^−1^, while the maximum allowed amount is 25 mg kg^−1^ [41]. In China, amounts of Ni in soils have been measured in the range of 40⎼200 mg kg^−1^, which was higher (up to three times) than the permissible amount of 60 mg kg^−1^ [42]. Similarly high levels can be observed in many other countries [43,44,45,46].

## 3. Impacts of Heavy Metal Pollution on Environment and Human Health

A major problem with heavy metal contamination is that their ions are not biodegradable, which causes them to circulate in the environment. Hence, they may persist in the environment in a toxic form for at least 200 years [1,3,4,5,6,14,15]. These soil-polluting compounds can inhibit the growth of soil microorganisms. They also lead to the disruption of the physiological functions of microorganisms, as well as disrupting processes related to the decomposition and transformation of the organic matter [47]. Disruption of the decomposition of organic matter by microorganisms can lead to an increase in the pool of bioavailable forms of metals in the soil. Importantly, the forms of heavy metals occurring in soil are one of the factors determining their mobility and toxicity in the environment [6,16,47].

In biological systems, heavy metals contribute to the interference of enzymatic processes, disruption of the function of subcellular structures and can cause damage through free radical processes, through physicochemical properties similar to metals that are physiologically active [5,45,48]. For instance, in the cell cytoplasm, metal ions readily bind to functional groups such as -SH, -OH, and -NH, which causes the protein molecule deformation and leads to a complete decrease in the biological activity of proteins, and consequently cell death [15,49].

The greatest risk of heavy metals results directly from their transport along the food chain and the phenomenon of bioaccumulation: the largest amounts of a substance are delivered to the last link, which is human (Figure 1) [6,8,48,49]. The primary source of human health exposure to heavy metals is food, mainly of plant origin [6,7,49]. Therefore, the accumulation of heavy metals in crops for animal feed and direct human consumption should be limited [6,49].

In the human body, heavy metals can cause acute poisoning and chronic conditions. Most of the symptoms of heavy metal poisoning do not become apparent immediately after onset, but after many months or even years have passed [5,50,51]. The spectrum of the toxicity of heavy metals in the human body is very wide [5]. Despite a similar mechanism of toxicity, individual heavy metals often tend to affect different tissues and organs [5,49]. Lead accumulates primarily in bone tissue, cadmium in renal cortical tissue and liver, while mercury accumulates in the form of methylmercury compounds in brain tissue, which can lead to severe neurological changes [5,6,7,49,50]. Moreover, some heavy metals and their compounds are classified as confirmed or probable carcinogens by the International Agency for Research on Cancer (IARC) [5,6,51]. In terms of the environmental risk, two elements have ranked first for years: Cd and Pb. They are followed by As, Cr, Hg, and Zn [1]. According to the IARC study, Cd is classified in Group 1, which includes substances that are carcinogenic to humans, while inorganic Pb compounds are classified in Group 2A, which includes substances that are probably carcinogenic to humans [6,51,52].

## 4. Heavy Metal Bioremediation Strategies Detected in *Bacillus*

Microorganisms can use several strategies to remove heavy metals present in the environment (Figure 2) [8,20,53,54]. Biosorption, bioaccumulation, and bioprecipitation are the most common heavy metal removal strategies of the genus *Bacillus* [8,55].

### 4.1. Biosorption

Biosorption is a physicochemical, metabolism-independent heavy metal uptake process based on cell membranes. It functions through compounds with a negative charge that are present in the cell membranes. Importantly, the biomass used for biosorption is usually non-living biomass, as this way, the process proceeds more efficiently than with living microorganisms. The efficiency of this strategy mainly depends on several parameters, including surface properties, e.g., functional groups present on the cell membrane, pH, temperature, or electrostatic interactions [12,57,58]. Understanding the biosorption mechanisms that enable the removal of heavy metals is crucial to optimizing the process. To date, several mechanisms occurring during the sorption process have been discovered, and different mechanisms can proceed at the same time at different rates. Among the biosorption mechanisms, the following mechanisms can be identified: (i) ion change, a reversible chemical reaction involving the exchange of ions for other ions of the same charge; (ii) complexation, heavy metal ions bind to functional groups present in cell membranes; (iii) physical adsorption caused by intermolecular interactions, including Van der Waals forces (Figure 2) [57,59].

To date, several papers on biosorption with *Bacillus* spp. have been published [21,60,61,62]. Some strains of the *Bacillus* spp. may also have the ability to bio-sorb a few different heavy metals. *B. thuringiensis* OSM29, isolated from the rhizosphere of cauliflower grown in soil irrigated with industrial effluent, was capable of remediating Cd, Cu, Cr, Ni, and Pb [63]. The biosorption capacity of *B. thuringiensis* OSM29 was highest for Ni (94%), while the lowest biosorption by the bacterial biomass was noted for Cd (87,0%). The researchers also observed that the biosorption efficiency was dependent on a few physicochemical parameters, such as pH, initial metal concentration, and contact time. For example, the optimum pH values for copper and lead biosorption efficiency was 6.0, while for Ni and Cr, it was 7.0. Additionally, using FTIR, the authors identified the following chemical functional groups in the studied strain: amino, carboxyl, hydroxyl, and carbonyl groups, involved in the sorption of heavy metals [63]. Nevertheless, most studies concern the biosorption of single heavy metals. Strains of the genus *Bacillus* are capable of Hg biosorption. For instance, Sinha et al. [64] analyzed a biosorption potential of immobilized *B. cereus* cells for bioremediation of mercury from synthetic effluent. Importantly, the experiment was conducted under various conditions. The maximum adsorption capacity of *B. cereus* (immobilized cells) was 104.1 mg g^−1^ (Hg^2+^), and was noted for a pH of 7.0 at 30 °C, for a pH of 7.0 after 72 h from contact, and biomass concentration of 0.02 g L^−1^. Moreover, the average free energy value calculated using the Dubinin–Radushkevich (D–R) model was 15.8 kJ mol^−1^, indicating that this process was chemically more favorable than the physical adsorption process [64]. Chen et al. [65] conducted a study on Pb(II) biosorption, using the strain *B. thuringiensis* 016 through batch and microscopic experiments. The authors noted that the highest biosorption potential of Pb for *B. thuringiensis* 016 was approximately 165 mg g^−1^ (dry weight). Interestingly, this study showed that pH, amide, carboxyl, and phosphate functional groups of the studied strain (studied by fourier transform infrared (FTIR) analyses and selective passivation experiments) greatly affected the Pb biosorption. Furthermore, the observation by scanning electron microscopy proved that Pb precipitates had accumulated on the surfaces of the bacteria cells [65].

Moreover, *Bacillus* spp. strains are also capable of biosorption of less toxic metals than those above. For instance, *B. cereus* AUMC B52 was capable of Zn biosorbing. The maximum adsorption capacity of *B. cereus* AUMC B52 calculated from the Langmuir adsorption isotherm was 66.6 mg g^−1^. In addition, the presence of amine, hydroxyl, carboxyl, and carbonyl groups, which are probably responsible for Zn(II) biosorption, was detected in the bacterial biomass using FTIR [66]. There are also examples of As biosorption by *B. cereus*, which is more often found in the environment in the form of anions. Giri et al. [67] detected an adsorption capacity of approximately 32 mg g^−1^ for arsenite at pH 7.5, at a biomass dose of 6 g L^−1^. The ability to bioabsorb arsenic was also noted in *B. thuringiensis* WS3. The maximum As(III) adsorption capacity was approximately 11 mg g^−1^, in the optimum As(III) removal conditions: 6 ppm As(III) concentration, pH 7, temperature 37 °C, and biomass dose of 0.50 mg ml^−1^ [68].

### 4.2. Bioremediation by Extracellular Polymeric Substances (EPS)

Another important mechanism for bioremediation of heavy metals that many metal-tolerant bacteria possess is the uptake of metals through the secretion of extracellular polymeric substances (EPS) (Figure 2) [58,69]. The EPS include compounds such as nucleic acids, humic acids, proteins, and polysaccharides, that bind cationic metals with varying degrees of specificity and affinity [58,70]. Their importance in the bioremediation process is based on their participation in the flocculation and the binding of metal ions from solutions [71]. Microorganisms that secrete exopolysaccharides are the most significant in the bioremediation of heavy metals [69]. Factors modulating the removal of metals by EPS include initial metal concentrations and pH [58].

To date, the ability to secrete EPS has been detected in several strains belonging to the genus *Bacillus*. For instance, multi-metal resistance (Pb, Cd, Cu, and Zn) strain, *B. cereus* KMS3-1, was able to produce EPS (optimum conditions: pH 7.0, 120 h incubation time, sucrose concentration 5 g L^−1^, and 10 g L^−1^ yeast extract) [72]. Furthermore, optimization of EPS production, using a central composite design, revealed that the optimal sucrose and yeast extract concentrations for enhanced EPS production (8.9 g L^−1^), were 5 g L^−1^ and 30 g L^−1^, respectively. In addition, using FTIR, thin layer chromatography (TLC), and high-performance liquid chromatography (HPLC) techniques, the researchers determined studied EPS as heteropolysaccharide, which consisted of glucose, mannose, xylose, and rhamnose [72]. However, there are also studies focusing on bioremediation using EPS against a single heavy metal contamination. Kalpan et al. [73] isolated an exopolysaccharide-producing bacteria, *B. cereus* VK1. Subsequently, EPS was purified, estimated, and further characterized by FTIR, gas chromatography, mass spectrometry (GC-MS), and thermo gravimetric analysis (TGA). Interestingly, using statistical modeling (response surface methodology, RSM), the researchers carried out media optimization to increase EPS production. The study results showed that *B. cereus* VK1 cultured in LB was capable of adsorbing up to 80.22 μg Hg^2+^ in 20 min, while the strain grown in RSM-optimized medium adsorbed up to 295.53 μg Hg^2+^ [73]. Moreover, the ability to produce EPS has also been detected in a species related to *Bacillus* spp., *Paenibacillus jamilae*. EPS showed a notable affinity for Pb in comparison to the other five metals. The bioremediation of lead (303.03 mg g^−1^) was as much as ten times higher than the removal of the other metals. Alongside that, studied EPS consisted of glucose (the most abundant sugar), rhamnose, galactose, fucose, and mannose [74].

### 4.3. Bioaccumulation

In contrast to the biosorption described above, bioaccumulation is a cellular energy-dependent process conducted by active metabolic microorganisms (Figure 2) [75]. Therefore, compared to biosorption, heavy metal uptake takes longer because it depends on the biochemical features, microbial internal structure, genetic and physiological ability, and environmental conditions affected by bioaccumulation activity [53,76]. Moreover, the bioaccumulation process was also found to be influenced by cell surface properties, including changes in charge. In addition, temperature also affects the bioaccumulation process: a higher temperature may significantly disrupt the metabolic activity of a bacterial cell [53,77]. The best-known mechanism of bioaccumulation is likely based on heavy metal binding using metallothioneins. Metallothioneins are cysteine-rich proteins (low molecular weight molecules, can be encoded by the *bmtA* gene that facilitates the bioaccumulation of heavy metals (e.g., Pb, Hg, Ni, Cd)) inside the cell [78]. Bacterial cells usually produce metallothioneins in reaction to enhanced exposure to metals [79,80]. This mechanism may be transferred by plasmids, facilitating its dispersion from one bacterial cell to another [81]. However, there are other bioaccumulation mechanisms that are often not universal for the bioremediation of all heavy metals. For instance, bioremediation of As in bacteria of the genus *Bacillus* is mediated by the ars operon through the use of the following genes: *arsA*, *arsB*, *arsC*, *arsD*, and *arsR*, where, e.g., the *arsA* and *arsB* genes have ATPase activity and *arsC* encodes an arsenate reductase that converts As(III) to As(V) (less toxic form) [82]. In contrast, Pb bioaccumulation in *Bacillus* spp. Is based on the *pbrD* gene [83], while the mechanisms leading to Cu accumulation by bacterial cells are encoded in the *cusF* gene, which contributes to the binding of copper in the periplasmic space [84]. In the case of mercury, bioaccumulation is related to the *merC* gene expression [85].

There are many studies confirming the ability of *Bacillus* spp. and related bacteria to bioaccumulate heavy metals [18]. For instance, metallothionein production has been detected in *Bacillus* spp., for example *B. cereus* and *B. megaterium* [86]. Importantly, bacteria of the genus *Bacillus* are capable of bioaccumulating various heavy metals. *B. cereus* RC-1, growing under various pH values and initial metal concentration, was able to remove a few heavy metals, such as Cu^2+^ (16.7% maximum removal efficiency), Zn^2+^ (38.3%), Cd^2+^ (81.4%), and Pb^2+^ (40.3%), with initial concentrations of 10 mg L^−1^, at pH 7.0 [87]. Interestingly, the bio-removal of the two crucial metals—Cd^2+^ and Pb^2+^—was paralleled by cellular uptake of Na^+^ and Mg^2+^ from the medium, respectively [87]. Alongside that, *B. coagulans* tolerated up to 512 ppm Cr(VI) concentration and had an MIC (minimum inhibitory concentration) of 128 ppm for Pb(II). Moreover, after 72 h, this strain had reduced 32 ppm Cr(VI) by 93%, and 64 ppm Pb(II) by 89.0% [88]. In addition, *B. cereus* BPS-9 has shown great potential of Pb accumulation (79.3%) [53]. The authors also found that, despite a reduction in the growth rate, the superoxide dismutase activity of *B. cereus* BPS-9 increased with increasing lead concentration, manifested by an increase in nitro blue tetrazolium (NBT) reduction from approximately 4% to 78% [53]. Moreover, bacteria of the genus *Bacillus* have been shown to have the ability to bioaccumulate the highly toxic arsenic for humans as well. For example, Singh et al. [89] detected the ability to bioaccumulate and volatilize As(V) in cultures of *B. aryabhattai*. The possibility of bioaccumulation of slightly less toxic Ni has also been recorded in bacteria of the species *B. cereus*. Naskar et al. [90] found that growing *B. cereus* M161 cells, depending on the growth phase of the culture, accumulated Ni(II) from the aqueous solution up to 80%, with surface binding (approximately 60%) dominating over intracellular accumulation (approximately 20%). On the other hand, the highest Ni(II) accumulation was recorded at 6.5, 32.5 °C, 2.5% inoculum volume, and 50 mL medium volume. However, no growth of the studied strain was observed at Ni ion concentrations beyond 50 mg L^−1^ [90].

### 4.4. Bioprecipitation

Bioprecipitation is another bioremediation strategy that has been found in bacteria. This strategy involves converting the concentration of free metals to insoluble complexes, thereby reducing their bioavailability and toxicity. Microorganisms can facilitate precipitation via catalyzing oxidative and reductive processes, leading to the precipitation of contaminants including Pb, Cd, Cr, Fe, and U. In some microorganisms, it has also been discovered that they can release phosphates and increase the precipitation of metal phosphates, while other bacteria are capable of precipitating hydroxides or carbonates by forming alkanes (Figure 2) [91]. There are not a lot of studies about precipitation carried out by the *Bacillus* spp. Nevertheless, bacteria of the genus *Bacillus* can bio-precipitate the most toxic heavy metals, including lead and cadmium. For instance, the lead-resistant strains—B. iodinium GP13 and B. pumilus S3—were found to facilitate the precipitation of lead in the form of lead sulphide (PbS) [92]. Moreover, bacteria capable of precipitating lead into lead phosphate (Pb_3_(PO_4_)_2_) also include *B. thuringiensis* 016 [65]. Another example of bioprecipitation by the *Bacillus* spp. is the study by Li et al. [93]: using analyses of energy dispersive spectroscopy, X-ray photoelectron spectroscopy, and select area electron diffraction, the authors showed that *B. cereus* Cd01 was capable of Cd bioprecipitation into polycrystalline and/or amorphous cadmium phosphate and cadmium sulfide. Furthermore, Molokwane et al. [94] observed Cr(VI) reduction by precipitation after the application was enriched by a mixed culture of bacteria consisting of bacteria of the genus *Bacillus*, including *B. cereus*, *B. thuringiensis*, and related genera, such as *Paenibacillus* and *Oceanobacillus*. The highest reduction of Cr(VI) in aerobic cultures was obtained at a high concentration of 200 mg L^−1^, after incubation for 65 h [94]. To our knowledge, there have been no studies to date describing the possibility of bioprecipitation of other heavy metals (important from the point of view of pollution) by bacteria of the genus *Bacillus*.

To summarize these subsections, it is worth adding that biologically enhanced precipitation may be used to remove metals and metalloids from a range of wastewaters, for example, acid mine drainage, electroplating, and tannery effluents [91].

### 4.5. Biological Removal of Heavy Metals Using Plant Growth-Promoting Bacteria

There are still not a lot of studies on the bioremediation by *Bacillus* spp. in terms of application perspective; most studies focus on bioremediation mechanisms and study bioremediation efficiency in aqueous solutions with heavy metals [65,90,94]. Furthermore, only a few studies present results on the bioremediation activity of *Bacillus* spp. without the involvement of plants. For instance, for the bioremediation of cadmium, a combination of the bacteria *B. megaterium* with earthworms (*Eisenia fetida*) was used. According to the experiment with Cd-contaminated soil (Cd at approximately 2.5 mg kg^−1^), this combination was more effective than bioremediation using only earthworms [95]. On the other hand, the vast majority of studies on the application of *Bacillus* spp. as bioremediation agents are also related to phytoremediation [96], indicating that the bioremediation action of *Bacillus* spp. is not limited to playing a role in the geochemical cycle of heavy metals in soil [97,98]. Moreover, heavy-metal-accumulating plants supported by bacteria of this genus may be used to produce biogas, and the digestate meeting the criteria for heavy metal content can be used as fertilizer. Thus, this type of approach appears to be the most appropriate in the context of bioremediation involving this microbial group [99,100].

Metal-accumulating plants can be enhanced by metal-resistant plant growth-promoting bacteria (PGPB), which can increase the efficiency of bioremediation [56,101,102]. Therefore, the use of PGPB has recently been expanded to include the potential remediation of contaminated soils with crops, energy plants, and hyperaccumulators (plants capable of accumulating extremely large amounts of heavy metals in their aboveground parts, without suffering from phytotoxic effects) [100,103]. Therefore, the application of PGPB has recently been expanded to include remediation of contaminated soils in combination with plant hyperaccumulators, that is, plants capable of accumulating extremely large amounts of heavy metals in their aboveground parts, without suffering from phytotoxic effects. Plant stimulation by PGPB has been observed by many authors [11,101,104,105,106,107]. PGPB may enhance plant growth either directly or indirectly. Mechanisms of direct action include production of various biological substances, for instance, indole-3-acetic acid (IAA), gibberellins, cytokinins, and 1-aminocyclopropane-1-carboxylic acid (ACC) deaminase, and atmospheric nitrogen fixation (nitrogenase production), or phosphorus solubilization [56,108,109,110,111,112]. On the other hand, indirect mechanisms include the production of antibiotics (for example, cyclic lipopeptides), enzymes such as chitinases, cellulases, and glucanases, and siderophore production [105,113,114,115,116].

The abilities that promote PGPB in phytoremediation processes include alleviating harmful effects caused by heavy metal pollution (e.g., reduced chlorophyll level and oxidative stress), boosting heavy metal tolerance of plants, and enhancing the accumulation of heavy metals in plant tissues [11,117,118,119,120,121]. Thus, bacteria can facilitate heavy metal remediation through several mechanisms. For instance, phytohormones, such as IAA, causing root elongation and surface area (enhancing nutrients uptake), lead to an increase in the plants biomass, which results in a larger phytoremediation surface area of plants [99,122,123]. In addition, beneficial microorganisms may help reduce ethylene stress in plants growing in metal-contaminated soil through the deaminase ACC activity, which breaks down the ethylene precursor, ACC. It results in the development of longer roots, thus enabling the phytoremediation process to proceed more efficiently [56,94,124,125]. Plant stress due to the presence of heavy metals can also be alleviated by the secretion of antioxidant enzymes by PGPB [96]. Additionally, PGPB releases siderophores, iron-chelating compounds that enhance iron uptake by plant roots in hostile, metal-contaminated environments [104,126,127,128]. Siderophores can also mobilize heavy metals, increasing metal accumulation by resistant bacteria (Figure 2) [56,129,130,131]. In turn, plant endophytes (e.g., root endosphere endophytes) can also enhance the phytoremediation through bioaccumulation mechanisms [122].

So far, several studies have been reported describing the support of plant phytoremediation by plant growth-promoting bacteria of *Bacillus* spp. [11,132,133,134]. Most of the research on this topic concerns experiments conducted under controlled conditions. For instance, a study conducted under gnotobiotic conditions showed the possibility of phytoextraction of cadmium- and lead-contaminated soils with bacteria of the genus *Bacillus* [135]. A heavy-metal-resistant, tomato growth-promoting strain *Bacillus* sp. RJ16 (Table 1) (which synthesized IAA, siderophores, and ACC deaminase to stimulate tomato root growth) led to an increase in Cd and Pb content in aboveground tissues from 92% to 113% and from 73% to 79%, respectively, in inoculated plants growing in heavy-metal-contaminated soil, compared to the control without inoculation [135]. Similarly, *B. subtilis* and B. pumilus were also able to facilitate the accumulation of various heavy metals, such as Cu, Cr, Pb, and Zn in tissues of Zea mays and Sorghum bicolor (greenhouse illuminated with natural light; total concentrations of heavy metals in soil: Cu 22,800, Cr 16,865, Pb 1900, Zn 32,500 mg kg^−1^ dry soil) [136]. Different patterns were noted by Saran et al. [137], who showed that after 2 months, sunflower (*Helianthus annuus*) seedlings grown on contaminated soil (Cd 0.42, Cu 1.02, Pb 5.48, Zn 12 mg kg^−1^) and inoculated with a plant growth-promoting strain B. proteolyticus ST89 (Table 1) achieved a 40% higher biomass production than uninoculated control plants, and accumulated 20% less Pb and 40% less Cd in aboveground plant parts, which indicates a reduction in phytotoxicity (controlled conditions, greenhouse) [137]. Interestingly, B. paramycoides ST9 (Table 1) increased the bioaccumulation factor of Pb three times and Cd six times, without suppressing plant growth [137]. However, most studies on this issue regarded the Cd bioremediation only. For instance, the application of rhizobacteria *B. subtilis* contributed to the reduction of the Cd bioavailability by approximately 39% in soil planted with ryegrass ( *Lolium multiflorum* L.), and enhanced Cd accumulation in ryegrass by nearly 28%. Moreover, the inoculation of this strain increased plant antioxidant enzymes and enhanced biomass by nearly 21% [96]. Additionally, using 16S rRNA sequencing, the researchers also assessed the change in the native microbiota following the introduction of PGPB. The study found that the application of *B. subtilis* in the rhizosphere microbiota caused significant changes, e.g., enrichment of the population of phylum Proteobacteria [96], which includes bacteria of the genus Pseudomonas. An increase in the abundance of this genus could also enhance bioremediation efficiency [138]. Furthermore, for Cd bioremediation, bacteria of the *Bacillus* genus could also be used with other bacteria. For instance, application of B. mycoides and *Micrococcus roseus* strains in Cd-contaminated soil (100 and 200 mg Cd kg^−1^) planted with maize (greenhouse experiment) contributed to increased Cd uptake in shoots and roots compared to the control [139]. Another example of consortium with *Bacillus* sp. application was the study by Pinter et al. [140]: the authors used a consortium that consisted of *B. licheniformis*, *Micrococcus luteus*, and *Pseudomonas* fluorescens, which increased the concentration of As(III) in the leaves, and increased the plant defense mechanisms which helped reduce the toxic effects of As(III). There are also cases of bioremediation of *Bacillus* spp. strains with plants of less toxic metals than those mentioned above. For instance, He et al. [141] documented that strains *B. subtilis* and *B. cereus* significantly enhanced the accumulation of Zn and the shoot and root biomass compared to non-inoculated plants in experiments conducted on *Orychophragmus violaceus* (greenhouse under controlled climatic conditions). In addition, it has also been shown that members of the genus *Bacillus* can assist in the phytoremediation of nickel and promote plant growth in nickel-contaminated soils. The inoculation of B. juncea by rhizobacteria *B. cereus* SRA10 (Table 1) also contributed to a notably enhanced growth of root and shoot Ni accumulation (greenhouse conditions) [142]. A slightly different pattern was noted by Rajkumar et al. [143] in a study conducted on B. juncea, which showed that *Bacillus* sp. Ba32 (Table 1), capable of producing siderophores and solubilizing phosphate, could stimulate the growth of this plant under Cr contamination, but did not affect the amount of chromium Cr accumulated in the roots and shoots (growth chamber conditions) [143].

Regarding the above, the majority of bioremediation experiments involving *Bacillus* spp. have been carried out under simple or controlled conditions, including growth chamber and greenhouse studies. However, studies on the effectiveness of *Bacillus* spp. in bioremediation have also been conducted under outdoor conditions. Sheng et al. [144] carried out an outdoor pot experiment, which demonstrated that soil inoculation with biosurfactant-producing *Bacillus* sp. J119 (Table 1) significantly increased tomato plant biomass and Cd uptake in plant tissues, thereby increasing the phytoextraction potential in soil contaminated with the aforementioned metal [144]. Moreover, Zaidi et al. [132] showed that the strain *B. subtilis* SJ-101 (Table 1) exhibited a protective activity against Ni phytotoxicity in Brassica juncea grown in soil treated with NiCl_2_, at concentrations ranging from 250 to 1750 mg kg^−1^ (pot experiments in open-field conditions). In addition, the study showed that the studied strain also had the ability to produce indoleacetic acid (IAA) and dissolve inorganic phosphate, which promotes the growth of the studied plant. Furthermore, the study indicated the possibility of using B. subtlis SJ-101 as a bacteria-assisted phytoaccumulation of this toxic heavy metal from polluted sites [132].

Importantly, by conducting research on the PGPB application for bioremediation of heavy metals under controlled conditions, researchers are reducing the number of factors affecting their effectiveness. As is well known, the effectiveness of PGPB is influenced by soil properties (including chemical and microbiological properties), which are modulated by a range of factors including meteorological conditions [116]. Thereby, there is still a great need for research conducted under field conditions, which will provide a broader view of the interactions between bacteria, plants, and soil, thus leading to an essential step in the transition from laboratory experiments to practical applications. Unfortunately, field trials using PGPB in phytoremediation are rarely reported [100,122]. An instance of such a study using *Bacillus* spp. is the experiment conducted by Wu et al. [122]. The authors revealed that *B. megaterium* BM18-2 (mutant) (Table 1) was able to increase Cd accumulation in the above-part of plants (hybrid *Pennisetum*) by nearly 29% (572 μg plant^−1^) compared to the control (pollutant concentration of cadmium was 0.50 mg kg^−1^).

## 5. Conclusions

Biological removal of toxic metals, using bacteria of the genus *Bacillus*, has now gained much interest in the context of bioremediation studies. Methods based on microorganisms including *Bacillus* spp. have several advantages over conventional physical and chemical techniques, including higher specificity, the possibility of using in situ, and ability of enhancing phytoremediation. In addition, the good adaptation of bacteria of this genus to unfavorable conditions and the fact that they produce spores provide a notable advantage over most bioremediation techniques. Importantly, the bioremediation efficiency using *Bacillus* spp. is constantly increasing. This phenomenon is generated by the use of statistical models to optimize bioremediation (in terms of conditions), the development of molecular techniques that will make it possible to use bacteria that show enhanced resistance to heavy metals in the future, and also application of PGPB as a phytoremediation-supporting agent, thereby increasing their potential in the bioremediation process. However, such solutions are still rarely translated into soil bioremediation, including the use of PGPB. Moreover, soil studies describing the possibility of using *Bacillus* spp. phytoremediation are generally conducted under controlled conditions. Changing the research approach by using bioremediation PGPB of the genus *Bacillus* more frequently in field conditions could contribute to the development of the research area, and consequently improve the application possibilities. Finally, it should also be mentioned that the development of next-generation sequencing (NGS), allowing more detailed insight into the crucial biodegradation pathways of these bacteria, provides an additional opportunity to enhance bioremediation by *Bacillus* spp. NGS techniques can also contribute to increasing the knowledge of the relationship between bacteria exhibiting bioremediation traits and the native microbial communities of contaminated environments. Such knowledge may lead to associations that will result in a synergistic cooperation.

## Figures and Tables

**Figure 1 ijerph-20-04964-f001:**
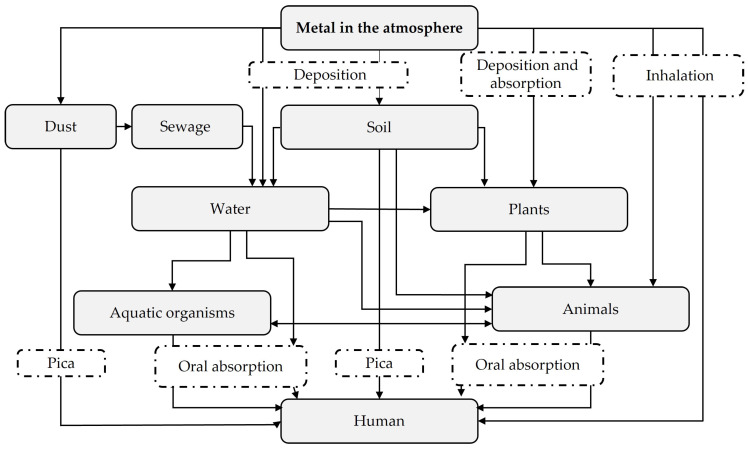
Diagram of environmental exposure and accumulation of metals in the human body, using lead as an example (according to Seńczuk [50]; modified).

**Figure 2 ijerph-20-04964-f002:**
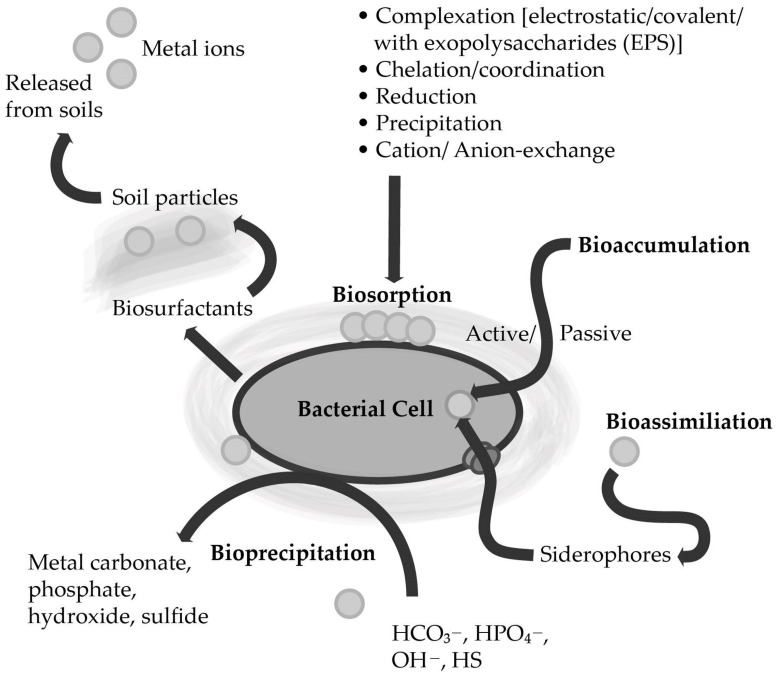
Various types of bacterial interactions with heavy metals accumulated in soils (according to Ahemad [56]; modified).

**Table 1 ijerph-20-04964-t001:** Phytoremediation supporting bacteria of the genus *Bacillus*, and their effects.

Strains	Plant	Bioremediated Metal	PGP Traits	PGP Effects	References
*Bacillus* sp. RJ16	*Solanum lycopersicum*	Cd and Pb	IAA, siderophores and ACC deaminase	Stimulatation of tomato root growth	He et al. [135]
*Bacillus cereus* SRA10	*Brassica juncea*	Ni	IAA, siderophores	Overall plant growth promotion	Ma et al. [142]
*Bacillus* sp. Ba32	*Brassica juncea*	Cr	Siderophores	Increase in root and shoot length	Rajkumar et al. [143]
*Bacillus proteolyticus* ST89	*Helianthus annuus*	Cd and Pb	IAA	Increase in biomass production	Saran et al. [137]
*Bacillusparamycoides* ST9	*Helianthus annuus*	Cd and Pb	⎼	Increase in shoot biomass production	Saran, et al. [137]
*Bacillus* sp. J119	*Brassica napus* Huiyou-50, *Zea mays* Denhai-11, *Sorghum bicolor × Sorghum sudanense*, *Lycopersicon esculentum* Shanghai-906	Cd, Pb, Zn and Cu	IAA	Increase in stem length	Sheng et al. [144]
*Bacillus subtilis* SJ-101	*Brassica juncea*	Ni	IAA	Increase in growth of above-ground tissue and root	Zaidi et al. [132]
*Bacillus megaterium* BM18-2	*Pennisetum americanum × Pennisetum purpureum Schumach*	Cd	IAA	Increase in shoot and root length	Wu et al. [122]

## Data Availability

The raw data supporting the conclusions of this article will be made available by the authors without undue reservation.

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
