# Peer review of "Bioremediation of Heavy Metals by the Genus Bacillus"

_ijerph, 2023, doi:10.3390/ijerph20064964_

Round 1

Reviewer 1 Report

Using bacteria to remediate heavy metal contaminated soil already became a promising measure with features of cost-effective, high efficiency and no second pollution. There is still a long way to go before its actual application in heavy metal remediation. Thus, summarizing the current application status and mechanism of bioremediation of heavy metal contaminated soil by bacteria could provide important basis for other researchers. Unfortunately, this manuscript lack innovation and couldn’t provide effective information for others thus needs major revise before publication. Please see detailed comments below:

1.       Line 24: Could you please clearly explicate which environment (water or soil) Bacillus has been used for heavy metal remediation? Actually this manuscript mainly focused on soil remediation.

2.       Introduction: Why this manuscript focuses on researches of using Bacillus to reduce heavy metal in soil? What are the unique characters of this genus comparing with other bacteria? The basis and significant of this review was not clearly stated.

3.       Line 72-101: When discussing the contamination scale of soil, do you mean farmland soil or other types of soil? The contamination situation, standard limits and remediation method varied a lot among different soil types. Thus it’s necessary to elucidate the application range of Bacillus. Also, the references are insufficient to support the argument, such as reference 43, only use one site research in industrial zone in Iran to represent the contamination situation in Europe; reference 14 also couldn’t represent the contamination situation in US.

4.       Line 102-Line159: I suggested shortening this part and briefly explaining the impacts of heavy metal contamination on environment and human being.

5.       I strongly recommend summarizing the application of bioremediation using Bacillus including the strain names, heavy metal contamination in soils, application approaches, remediation effects, etc.

6.       Remediating mechanisms is the most essential part of this manuscript, but as different metals had different properties such as arsenic usually stay as anion other than cation in soil, so they should be assumed different mechanisms and should be elucidated specifically according to their species.

7.       I also suggested summarizing in table about the biological removal of heavy metal using plant growth-promoting bacteria, including the strains, plant species, heavy metals, promoting effects, etc. Since the coordination between bacteria and plants is complicated, how does bacteria enhance the phytoremediation efficiency should be more specifically elucidated in this manuscript.

8.       Conclusion and future prospects: the conclusion is not concise enough to summarize the manuscript. In addition, no specific comments for future use of Bacillus are mentioned.

Author Response

Response to Reviewer 1 Comments

Point 1: Line 24: Could you please clearly explicate which environment (water or soil) Bacillus has been used for heavy metal remediation? Actually this manuscript mainly focused on soil remediation.

Response: We have completed the information that for the purposes of this paper we are focusing on soil remediation.

Point 2: Introduction: Why this manuscript focuses on researches of using Bacillus to reduce heavy metal in soil? What are the unique characters of this genus comparing with other bacteria? The basis and significant of this review was not clearly stated.

Response: The unique trait of Bacillus spp. is the ability to spore-forming under extreme conditions. Due to their specific structure, the spores are able to resist significant environmental stresses, including high temperature, drought, humidity and radiation; this characteristic gives them an advantage over other bacteria and makes them eagerly used commercially in various fields of industry and agriculture. We have also inserted this clarification in the text.

Point 3: Line 72-101: When discussing the contamination scale of soil, do you mean farmland soil or other types of soil? The contamination situation, standard limits and remediation method varied a lot among different soil types. Thus, it’s necessary to elucidate the application range of Bacillus. Also, the references are insufficient to support the argument, such as reference 43, only use one site research in industrial zone in Iran to represent the contamination situation in Europe; reference 14 also couldn’t represent the contamination situation in US.

Response: In discussing heavy metal contamination of soils, we have not focused on a specific type of soil, we present the overall scale of the problem. The references we used were incorrectly numbered, which is misleading. Thus, at least the data on the soil contamination situation in Europe was not taken from the study for Iran, so it was not based on a single country – the data was compiled and published by the European Environment Agency (EEA). The data collection covers 39 countries. The reference numbering has again been carefully checked. There was a numbering mistake in the list of references, but this has been corrected.

Point 4: Line 102-Line 159: I suggested shortening this part and briefly explaining the impacts of heavy metal contamination on environment and human being.

Response: We have shortened this part of the paper to discuss more briefly the impact of heavy metal pollution on the environment and humans.

Point 5: I strongly recommend summarizing the application of bioremediation using Bacillus including the strain names, heavy metal contamination in soils, application approaches, remediation effects, etc.

Point 7: I also suggested summarizing in table about the biological removal of heavy metal using plant growth-promoting bacteria, including the strains, plant species, heavy metals, promoting effects, etc. Since the coordination between bacteria and plants is complicated, how does bacteria enhance the phytoremediation efficiency should be more specifically elucidated in this manuscript.

Response: We have made a number of changes to this aspect: We have combined the application issue of bioremediation with phytoremediation with PGPB in the text - because most of the articles describe bioremediation with PGPB. We have also added new examples which describe it according to your recommendations. Besides, we have drawn up a summary table of bioremediation with PGPB while focusing on the plant growth promotion aspects in order not to duplicate the content related to the effects of bioremediation which is described in the manuscript. In addition, we have explained in more detail the relation between PGPB and phytoremediation plants.

Point 6: Remediating mechanisms is the most essential part of this manuscript, but as different metals had different properties such as arsenic usually stay as anion other than cation in soil, so they should be assumed different mechanisms and should be elucidated specifically according to their species.

Response: Charge is important, for example, for biosorption (by Bacillus spp.) and also bioaccumulation involving metallothioneins (cations only). There are few publications on As biosorption. Nevertheless, we found studies that show the possibility of As biosorption and bioaccumulation and have included them in the manuscript.

Point 8: Conclusion and future prospects: the conclusion is not concise enough to summarize the manuscript. In addition, no specific comments for future use of Bacillus are mentioned.

Response: In order to improve our conclusions, we have introduced additional information on the development of bioremediation research under field conditions.

Reviewer 2 Report

The authors present an interesting review with focus on the bioremediation of heavy metals by the genus Bacillus, which well summarized the current advance in this topic. Overall, the manuscript was well written. I recommend its acceptance after minor revisions.

I have two major suggestions. First, heavy metals are very common pollutants, which were widely recognized. The introduction section can be more concise. Second, many citations do not match the references, which should be checked carefully.     

L25: “eco-friendly” is an adjective, and should be replaced.

L27, L102: subtitle should be numbered.

L28-37: I don’t think this paragraph is necessary, because this is common knowledge.

L52: Check [13] is correct or not. I doubt the citation is [12] (L428-429). The citations should be carefully checked and confirmed they match the references well.

L53: [14] may be [13]

L295: Cr6+ may be a wrong spelling; it is better to write it as Cr(VI) or Cr(+6)

L339: italicize “Sorghum bicolor”

L425-427: This reference was wrong numbered.  

Author Response

Response to Reviewer 2 Comments

Point 1: L25: “eco-friendly” is an adjective and should be replaced.

Response: OK, we have changed the term to “sustainable environmental management”.

Point 2: L27, L102: subtitle should be numbered.

Response: All subtitles have been numbered.

Point 3: L28-37: I don’t think this paragraph is necessary, because this is common knowledge.

Response: This paragraph has been deleted.

Point 4: L52: Check [13] is correct or not. I doubt the citation is [12] (L428-429). The citations should be carefully checked and confirmed they match the references well.

Point 5: L53: [14] may be [13]

Response: Reference numbering has been carefully checked. There was a numbering mistake in the list of references, but this has been corrected.

Point 6: L295: Cr6+ may be a wrong spelling; it is better to write it as Cr(VI) or Cr(+6)

Response: Cr6+ was converted to Cr(VI).

Point 7: L339: italicize “Sorghum bicolor.”

Response: "Sorghum bicolor" has been italicized.

Point 8: L425-427: This reference was wrong numbered.

Response: Reference numbering has been carefully checked. There was a numbering mistake in the list of references, but this has been corrected.

Reviewer 3 Report

This is an excellent review on the sue of Bacillus species for remediation of metal contaminated sites. It is well written and organized and more important the authors have interpreted the literature for use by others. I found a few mistakes in English which I show in track changes in the attached WORD document.

Author Response

Response to Reviewer 3 Comments

Response: Thank you for the favorable review, we have incorporated the comments into the text.

Reviewer 4 Report

The article entitled "Bioremediation of heavy metals by the genus Bacillus" makes a bibliographic review on the state of the art in relation to the different mechanisms that some Bacillus genus species use to mobilize heavy metals in bioremediation processes.

The topic addressed is of scientific interest and responds to a current and growing problem. However, I consider that there is a lack of systematic approach. Here are some general considerations:

1. The authors refer to the concept of heavy metals, but they do not give the same weight to all of them. For the review to be complete and systematic, all of them should be addressed under each of the headings. In the event that there are no reviews on a specific heavy metal for any of the mechanisms, mention it explicitly. This would make reading tracking easier and more useful as a reference tool.

More specifically:

1. An English review must be completed.

2. Some genera/species need to be italicized.

3. It is not common to find "of the genus Bacillus". I would recommend reviewing the entire MS and replacing with "Bacillus genus/ Bacillus spp.".

4. The collocation "interestingly" is repeated too frequently. Review all the MS.

5. L 22-23. The authors state that, for these reasons, the Bacillus genus is one of the best sustainable solutions to reduce heavy metals. This review focuses exclusively on this genre, with many others involved in this process. No evidence is provided to support such a conclusion. Removal or shading is recommended.

6. L 38-44. Reduce the paragraph, since it is information of a very basic nature.

7. L 71. The authors speak of phytoremediation, but have not previously explained the concept. In addition, whenever this concept is referred to with reference to heavy metal hyperaccumulator plants. This is not correct, or at least not accurate. Bioremediation processes do not have to include hyperaccumulators, they can simply be metal-tolerant plants that do not accumulate metals in their tissues (Lupinus sp., for example).

8. L 99-101. Eliminate.

9. L 105-106. These contaminants inhibit the growth of soil microorganisms. Not necessarily. Selection and speciation phenomena (cladogenesis and anagenesis) occur, among the most resistant. But not general inhibition. Review and correct, giving a more precise explanation.

10. L 127-153. Justify the text.

11. L 170-171. The biomass used in biosorption is non-living. This phrase is not well understood and is not well justified. Is it always dead biomass? including the genus Bacillus? doesn't it work with live microorganisms?

12. The authors focus on the origin of contamination by heavy metals, however, on many occasions natural sources are also important. At least the authors should mention it.

13. In general, review the entire MS to either always put the chemical elements in letter (Lead) or in symbol (Pb). Homogenize.

For all of the above, a Major review is recommended to the authors, especially for what is mentioned in general consideration 1.

Author Response

Response to Reviewer 4 Comments

The authors refer to the concept of heavy metals, but they do not give the same weight to all of them. For the review to be complete and systematic, all of them should be addressed under each of the headings. In the event that there are no reviews on a specific heavy metal for any of the mechanisms, mention it explicitly. This would make reading tracking easier and more useful as a reference tool.

Response: Our approach for this chapter is based on the description of the most important bioremediation strategies in Bacillus spp. Nevertheless, we have supplemented the information on bioaccumulation, bioprecipitation and bioprecipitation of the most important heavy metals, and signaled the lack of information in the case of bioprecipitation of some metals. In addition, we have also expanded the topic related to bioaccumulation mechanisms.

Point 1: An English review must be completed.

Response: The review in English has been completed.

Point 2: Some genera/species need to be italicized.

Response: All text was carefully checked, and words that required it were italicized.

Point 3: It is not common to find "of the genus Bacillus". I would recommend reviewing the entire MS and replacing with "Bacillus genus/ Bacillus spp.".

Response: In our opinion, the form 'genus Bacillus' is also frequently used. To confirm our opinion we have attached below links to the papers from leading microbiology journals.

https://www.nature.com/articles/s41586-018-0616-y

https://www.frontiersin.org/articles/10.3389/fmicb.2020.01782/full

https://www.frontiersin.org/articles/10.3389/fmicb.2020.00353/full

https://www.sciencedirect.com/science/article/pii/S0944501320300173

https://www.cell.com/trends/microbiology/fulltext/S0966-842X(19)30072-1?sf215663762=1

https://www.microbiologyresearch.org/content/journal/ijsem/10.1099/ijsem.0.003775

Point 4: The collocation "interestingly" is repeated too frequently. Review all the MS.

Response: We have removed a few "interestingly" from the text.

Point 5: L 22-23. The authors state that, for these reasons, the Bacillus genus is one of the best sustainable solutions to reduce heavy metals. This review focuses exclusively on this genre, with many others involved in this process. No evidence is provided to support such a conclusion. Removal or shading is recommended.

Response: The unique trait of Bacillus spp. is the ability to spore-forming under extreme conditions. Due to their specific structure, the spores are able to resist significant environmental stresses, including high temperature, drought, humidity and radiation; this characteristic gives them an advantage over other bacteria and makes them eagerly used commercially in various fields of industry and agriculture. We have also inserted this clarification in the text.

Point 6: L 38-44. Reduce the paragraph, since it is information of a very basic nature.

Response: This paragraph has been shortened.

Point 7: L 71. The authors speak of phytoremediation, but have not previously explained the concept. In addition, whenever this concept is referred to with reference to heavy metal hyperaccumulator plants. This is not correct, or at least not accurate. Bioremediation processes do not have to include hyperaccumulators, they can simply be metal-tolerant plants that do not accumulate metals in their tissues (Lupinus sp., for example).

Response: The text was supplemented with information explaining what phytoremediation and hyperaccumulators are. The comment about the possibility of using plants that are not hyperaccumulators for bioremediation is right. Nevertheless, it is the hyperaccumulators that most of this type of scientific research focuses on, and undoubtedly these plants show the greatest effectiveness.

Point 8: L 99-101. Eliminate.

Response: L 99-101 has been eliminated.

Point 9: L 105-106. These contaminants inhibit the growth of soil microorganisms. Not necessarily. Selection and speciation phenomena (cladogenesis and anagenesis) occur, among the most resistant. But not general inhibition. Review and correct, giving a more precise explanation.

Response: We have changed the meaning of the content to may inhibit microbial growth which is correct.

Point 10: L 127-153. Justify the text.

Response: The text in L 127-153 has been justified.

Point 11: L 170-171. The biomass used in biosorption is non-living. This phrase is not well understood and is not well justified. Is it always dead biomass? including the genus Bacillus? doesn't it work with live microorganisms?

Response: Biosorption is a metabolism-independent mechanism, in which case it proceeds faster than with living microorganisms. We have introduced this clarification in the text.

Point 12: The authors focus on the origin of contamination by heavy metals, however, on many occasions natural sources are also important. At least the authors should mention it.

Response: Information on natural sources of heavy metals has been added in the text.

Point 13: In general, review the entire MS to either always put the chemical elements in letter (Lead) or in symbol (Pb). Homogenize.

Response: The notation of chemical element names to their symbols has been homogenized.

Reviewer 5 Report

This article summarized the information about the remediation effect of Bacillus spp. in heavy metal contaminated soil. However it only lists and collects the functions of Bacillus, there is still a lack of in-depth thinking on its role in soil remediation and future development trend. 

Author Response

Response to Reviewer 5 Comments

This article summarized the information about the remediation effect of Bacillus spp. in heavy metal contaminated soil. However it only lists and collects the functions of Bacillus, there is still a lack of in-depth thinking on its role in soil remediation and future development trend.

Response: In order to improve our work, we have rebuilt the chapter "Biological removal of heavy metal using Plant Growth-Promoting Bacteria", including a divided of the content in terms of the conditions under which the experiments were conducted and, importantly, highlighted what are the gaps in the research and what positives may arise in the future from their completion. We have also described in more detail the impact of PGPB on the efficacy of plant bioremediation. In addition, we have introduced new information on bioaccumulation mechanisms and supplemented information on biosorption, bioaccumulation and bioprecipitation of important heavy metals. Besides, we have also made changes to the conclusions. 

Round 2

Reviewer 1 Report

Most of the questions have been well responded. There are still some places need to be revised.

1.       I suggested removing table 1 because this is no relevant to your paper, unless you discuss the bioremediation strategies of all the elements listed here. Also Be is not a heavy metal.

2.       As for Section 4 “Heavy metal bioremediation strategies”, there are many references piled up here and lacks logic. I suggested writing in the order of elements or in the order of bioremediation effect, mechanisms and influencing factors. There are also some spelling mistakes, such as 4.2 “absporing” should be “adsorbing”. The manuscript requires further editing.

3.       Section 4.1 and 4.3: Why biomass used for biosorption is usually non-living? Living biomass also could remove heavy metal through biosorption. Aslo, how could you distinguish bioaccumulation with biosorption?

4.       Since this manuscript mainly focused on bioremediation of heavy metal contaminated soil, however, only few researches have been done in the soil culture system. I suggested analyzing the possible reasons and giving suggestions on future research.

Author Response

Response to Reviewer 1 Comments

Point 1: I suggested removing table 1 because this is no relevant to your paper, unless you discuss the bioremediation strategies of all the elements listed here. Also Be is not a heavy metal.
Response: In accordance with the reviewer's comment, we resigned from table 1.

Point 2: As for Section 4 “Heavy metal bioremediation strategies”, there are many references piled up here and lacks logic. I suggested writing in the order of elements or in the order of bioremediation effect, mechanisms and influencing factors. There are also some spelling mistakes, such as 4.2 “absporing” should be “adsorbing”. The manuscript requires further editing.

Response:  We have arranged our text according to the toxicity of the individual metals.

Point 3: Section 4.1 and 4.3: Why biomass used for biosorption is usually non-living? Living biomass also could remove heavy metal through biosorption. Also, how could you distinguish bioaccumulation with biosorption?

Response: Biosorption is a metabolism-independent mechanism and the biomass used for biosorption is usually non-living biomass, as this way the process proceeds more efficiently than with living microorganisms (can also be carried out using biomaterials, e.g. chitin and chitosan). In addition, biosorption is concerned with the surface binding of metal ions due to the functional groups including amine, hydroxyl, carboxyl, and carbonyl groups. In contrast, bioaccumulation is associated with a number of substances dependent on the expression of various genes, e.g. those responsible for the production of metallothioneins.

Point 4: Since this manuscript mainly focused on bioremediation of heavy metal contaminated soil, however, only few researches have been done in the soil culture system. I suggested analyzing the possible reasons and giving suggestions on future research.

Response: Bioremediation with the use of bacteria and plants is more efficient, which is probably why there is more studies on the subject. Besides plants supported by bacteria of this genus can be used to produce biogas and the digestate meeting the criteria for heavy metal content can be used as fertilizer. We have also inserted this clarification in the text.

Reviewer 4 Report

After reviewing the corrections of the authors, including the suggestions of the entire team of reviewers, the MS has improved substantially.

For this reason, I accept its publication in its current form.

Author Response

We thank you for your time and effort in reviewing our manuscript. The feedback has been invaluable in improving the content and presentation of the paper.